# God and Evil—Systematic-Theological Reflections on the Doctrine of God

Christian Danz 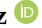

Protestant Theological Faculty, University of Vienna, 1010 Vienna, Austria; christian.danz@univie.ac.at

**Abstract:** Against the background of the current debates about God and evil, the article elaborates in three stages of argumentation the thesis that statements about God must not be understood as factual or representational statements, but as descriptive elements of the reflexive structure of the Christian religious communication. On this basis, a new perspective on God's relationship to evil in the world emerges, which, in contrast to the so-called theodicy debates, includes the self-view of the religious practitioners.

**Keywords:** theodicy; doctrine of God; religion; evil

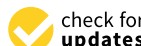



## 1. Introduction

"Either God wants to eliminate the evils and cannot, or he can and does not want to, or he cannot and does not want to, or he can and wants to. Now, if he wants to and cannot, he is weak, which is not true of God. If he can and does not want to, he is begrudging, which is also foreign to God. If he does not want to and cannot, then he is both begrudging and weak, and then also not God. However, if he wants to and can, which alone befits God, whence come the evils and why does he not take them away?" (Epicurus 1991, p. 136).

The question of how God is to be understood in the face of evils in the world was already a preoccupation of ancient philosophy, as Epicurus' considerations quoted show. In modern times, the name theodicy has become established for this task (cf. Leibniz 1996). Theodicy is concerned with an argumentative justification of God's goodness in the face of the objections raised by reason on account of the evils in the world.[1] The problem is evoked by the monotheistic idea of God as the creator of the world, as well as the determinations that belong to him. If God is perfectly good and at the same time omnipotent, how can there be evils in the world created by him? Against the background of these three statements—(a) God is perfectly good, (b) God is omnipotent, (c) there is evil in the world —the theodicy debate assumes the task of argumentatively demonstrating that either they can be true together or not. In this way, there arise justifications of God in the face of the evils in the world or denials of the existence of God or one of his so-called attributes, namely either his goodness (cf. Jordan 2020, pp. 273–86), his omnipotence,[2] or the evil.[3]

The following considerations are not intended to produce another positive or negative proposal how the three statements can or cannot exist together. Rather, it must be shown that a discussion of the relationship between God and evil, oriented towards the three statements, does not reach neither a plausible positive nor a negative result. The theoretical unanswerability of the theodicy problem, according to the thesis of the remarks, must be understood as an indication that statements about God are not supposed to be understood as factual or representational statements, but as descriptions of the reflexive structure of religious communication. For this reason, the considerations focus on the religious idea of God and its function in religion. An exhaustive treatment of the understanding of evil must therefore be deferred (cf. Dalferth 2006, 2008; Phillips 2005).

The structure of the following explanations results from the stated thesis that statements about God must be understood as descriptions of the reflexive structure of religious communication. We will begin with an overview of the current theodicy debates. It must be shown that these debates neither lead to a positive nor to a negative solution of the problem at issue. Against this background, the third section develops the proposal to understand God as a representation of the Christian religion. On this basis, the relationship between God and evil can then be described in the concluding fourth section within the framework of a theology of Christian religious communication.

## 2. Theodicy Discourses

The presupposition of a metaphysical-theistic idea of God, to whom the attributes of perfect goodness and omnipotence are ascribed, is fundamental to the current debates about the relationship between God and evil. On the condition that this God is at the same time the creator of the world in which, however, evil occurs, the question arises whether, and, if so, how, the statements about God's perfect goodness, his omnipotence, and the existence of evil in the world created by him can be true at the same time. By dealing with the possible proof that these three statements can exist together, the so-called theodicy problem takes on a logical form. However, the problem is intensified by two further additional assumptions, namely that (d) good is contraposed to evil and overcomes it and (e) omnipotence must be understood in the sense of boundlessness. Through these two additional assumptions, the three statements enter into a contradictory opposition (*kontradiktorischen Widerspruch*). In this form, John L. Mackie exposed the argument in his classical essay *Evil and Omnipotence*.[4] His evidential argument from evil is still in the background of the contemporary controversies about the logical compatibility of God's goodness and omnipotence with the existence of evil in the world (cf. Rowe 1979, pp. 335–41; Howard-Snyder 1996; Jordan 2020, pp. 275–77). In the following, Mackie's argument must first be briefly outlined. After this brief sketch, Richard Swinburne's argument that the (physical) evils of the world are compatible with the assumption of the probable existence of God will be presented as a counterpoint to Mackie's position. On the basis of Mackie's and Swinburne's alternative solutions to the problem of theodicy, the continuation of these positions in the contemporary debates can be examined and subjected to critical reflection.

Mackie's argument in his essay *Evil and Omnipotence* aims at proving that the three statements are in a contradictory opposition, i.e., they cannot be, in a necessary manner, true at the same time. This contradiction results, as noted above, from the assumption of the two additional premises.[5] If God is perfectly good and his omnipotence has no limits, then both statements cannot be true together if there is evil in the world. For God would be perfectly good and omnipotent only if there were no evils in the world he created. However, since there are evils, the statements that God is good and omnipotent cannot be true, given that if God were good and omnipotent, he would overcome evil. Even the acceptance of human freedom as one willed by God, which functions as the cause of evil, does not lead out of this dilemma. This hypothesis does not exonerate God from evil because it abolishes its omnipotence (cf. Mackie 1990, pp. 33–36). Thus, evil in the world falsifies the assumption of the existence of a good and omnipotent God. God is either good and not omnipotent, or omnipotent and not good.

Richard Swinburne has contradicted this conclusion. In his argumentation, he starts from similar premises as Mackie. However, Swinburne eliminates Mackie's two additional assumptions that drive the three statements into a contradictory opposition. As a result, Swinburne is able to hold to the probability of the existence of a good and omnipotent God despite the evils in the world. For Swinburne, similarly to Leibniz, physical evils, that is, *malum physicum*, are a necessary part of the world created by God and do not contradict the assumption of a good and morally acting God. There are, therefore, moral reasons that justify the admission of evil. Without (physical) evils, Swinburne argues, human beings would not be able to learn. Evils, then, have a necessary function for the experiential acquisition of knowledge.[6] They lead to a higher-order good, namely, the

knowledge to avoid evils and to act freely. If God's goal is to create free and responsible human beings, then he must necessarily allow the possibility of evils.[7] If this is true, then the three statements, God is perfectly good and omnipotent and there is evil in the world, do not contradict each other and can therefore be true at the same time (cf. Swinburne 1987, pp. 302–3).

With Mackie and Swinburne, the basic alternative of the theodicy debates is named. While for Mackie the three statements, God exists as a good as well as omnipotent being and there is evil in the world, cannot be true at the same time, for Swinburne they are. However, Swinburne's argument remains aporetic. Evils, in his conception, have a necessary function for the emergence of higher goods. However, if evils have the function of being means to the good, then they are themselves good. This means, however, that Swinburne's attempt to prove that the coexistence of the three propositions works only through a functional cancellation of evil: evil itself surreptitiously becomes good.[8] The problem just mentioned also confronts the continuations of Swinburne's argument in the contemporary controversies. By arguing that God has moral reasons for allowing evils in the world, one places evils in a superordinate context of meaning that necessarily tends to abolish evils.[9] Even an argumentation that posits God as strictly transcendent in the sense of a *potestas absoluta* and distinguishes his morality from that of the world does not develop beyond a functionalization of evils. By postulating God in this sense as absolute omnipotence in order to dissolve the evidential argument from evil, one dissolves the concept of God itself, since such a God can no longer be distinguished from the devil (cf. Jordan 2020, pp. 273–86).

Attempts to logically justify the coexistence of the three statements, God is good as well as omnipotent and there is evil in the world, lead, as we have seen, to a dissolution of the evil. With the goodness and omnipotence of God, the evils in the world created by him are only compatible if there is a moral reason for God to allow them. It is precisely this justification of the evils that functionalizes and thereby abolishes them.[10] However, justifications of God's goodness and omnipotence in the face of evil no longer differ from arguments that deny that all three statements can be true together. As we have seen, it was already Mackie's thesis that the propositions that God is good and omnipotent could not coexist with the proposition that there are evils in the world. Further development of his argument has confirmed this view. James P. Sterba has clarified in various publications that moral evils contradict the assumption of a morally good and omnipotent God (cf. Sterba 2018, pp. 173–91; 2019; 2020, pp. 203–8). His argument targets the moral reasons God might have for permitting evil and it works with the distinction between permitting and preventing. If God himself acts morally, Sterba argues, then he would have to prevent evils. However, since he does not, which is evident, then God is either not morally good or not omnipotent or both (cf. Sterba 2020, p. 208).

Both defenders of God's omnipotence and goodness in the face of the evils in the world and their opponents share the same presuppositions as well as the logical procedure. The starting point is a metaphysical theistic idea of God, from which statements are produced whose compatibility is demonstrated or disputed in a logical procedure. However, even if under different signs, all these attempts come to a similar result. They resolve surreptitiously at least one of the three statements—God is good, God is omnipotent, there is evil in the world created by him—to draw admittedly different conclusions. This result, however, indicates that the entire procedure, including its presuppositions, is problematic. Not only does it hide the self-view of the persons concerned[11] by treating the theodicy problem as a general logical problem, but it also claims the idea of God as a principle for explaining the world. Problems such as those just mentioned raise the question of whether the three statements that produce the theodicy problem can be understood as factual statements about God at all. However, that is not the case. Statements about God, this article proposes, must be understood as descriptions of the reflexive structure of religious communication. On this basis, as will be shown, a new perspective on the problem of God and evil emerges.

### 3. God in the Christian Religious Communication

As we have seen, the logical debates about the relation of God to evil start from a metaphysical theistic idea of God. Statements are produced about God as they are produced about an object, which must be true or false. However, the presupposition of a given or postulated God, to whom statements can be admitted or denied, is confronted with both epistemological and religious objections. Against the background of the modern critique of knowledge, every idea of God and every assertion of the reality of God is a human positing and thus can be annulled again. For a religious-philosophical or theological thematization of God, this means that it cannot begin with the assertion of God's existence. God is not an object that is somehow given, nor can he be derived from the world, as is consistently assumed in the theodicy debate. The world as such does not refer to God as its ground. Only in the Christian religion is God the creator of the world. However, in modernity, religion is a cultural form alongside other cultural forms. A theological doctrine of God, which takes into account the modern critique of knowledge as well as the differentiation of culture, must consequently begin with the concept of religion and address God as a component of religion. This procedure takes up and continues the development of modern Protestant theology since 1800, which distinguishes between theology and religion and, on the level of theological science, relates the Christian religion to an underlying concept of religion. Scientific theology no longer understands its contentual statements (*gegenständlichen Sinne*) in a representational sense, but as an expression and representation of religion.[12] On the basis of the distinction between theology and religion, scientific theology has the task to describe, in a methodically controllable way, how the contents of religion emerge together with them. God is consequently a component of the Christian religion, which is only given in it.

Religion, which here refers to the Christian one,[13] has become in the history of development of (Western) modernity a particular form of communication besides other forms in culture. Christian religion is autonomous when it is self-referential, that is, when religious communication refers exclusively to itself as religion. Consequently, the knowledge to communicate religion is also part of religion. The task of theology is to describe the inner functioning of the Christian religion from the self-view of those who practice it. Since theology is science and not itself religion, it can only construct the self-view of the Christian religion (cf. Danz 2021b, pp. 139–54). As a science (*Wissenschaft*), theology constructs in itself a complete image of the Christian religion by describing it as a self-referential and self-transparent communicative event that represents itself and its inner workings as religion in the idea of God. By referring to God, the Christian religion refers to itself and represents itself. In the considerations that follow, the systematic foundations of the concept of God in the Christian religion must be briefly outlined.

God and religion emerge simultaneously in and with the Christian religious communication. The classical justifications of the Christian religion in an already given religious object or in an already given religious subject are abandoned here. God and a religious subject are components of the Christian religion, but not presuppositions from which the Christian religion could be derived or justified. Rather, the Christian religion emerges from itself in the Christian religious communication. This is a tripartite interrelationship of content, appropriation, and articulation (cf. Danz 2019, pp. 118–30; Wittekind 2018, pp. 29–55). As a religion, Christianity is dependent on a determined contentual communication,[14] which must already exist as a distinct form of communication in culture. However, the Christian tradition handed down in culture is not yet itself religion, but merely a reference to religion. The handed-down communication becomes religion only when it is appropriated by people as Christian religion. The appropriation of the Christian religion forms a particular structural element, since it can be neither contained in the handed down contents nor derived from these contents. However, for the Christian religion to constitute itself as a religion, a third structural element must be added, namely the symbolic articulation of the appropriated Christian religious communication. Only when Christian religious communication is articulated and embodied does it become visible and exist in culture.

The Christian religion consists of the religious use of the appropriated Christian religious communication and it arises from all three structural elements together: It depends on certain contents that must be appropriated and articulated as religion. Apart from the religious use of content in the Christian religion, the Christian religion cannot exist at all. Therefore, it is not sufficient to limit oneself—as in the theodicy debates—only to the content level of religious statements. Contents such as God, God's omnipotence and goodness, etc., do not yet provide sufficient information as to whether they are intended to be used religiously or culturally. Religious content can also be used non-religiously in communication at any time, for instance philosophically, historically, aesthetically, etc. Consequently, in order to identify religion, the religious use and the religious intendedness of the contents in communication must be included in the determination of the concept of religion.

The Christian religion, as it has been shown, is a transparent, self-referential, and structured communication event. It presents itself with its contents and its functioning as religion. Its contents do not refer to objects given outside the communication, but to the communication itself. Christian religious contents have a reflexive function. They express in the Christian religious communication that these contents are intended as religion. This is how the function of the idea of God in the Christian religion is derived. By referring to God, the Christian religion refers to itself and presents itself as a transparent self-relation. God, as a representational content, describes the Christian religion itself both as an absolute self-relation and its knowledge of being religion. Religion and God are bound together here. When the Christian religious communication succeeds, that is, when it becomes real in culture as an autonomous form of communication, God comes into reality with it at the same time.[15] Since the Christian religion, by referring to God, represents itself, the Christian idea of God is to be understood *ab ovo* in a Trinitarian way. With God the Father, God the Son, and God the Holy Spirit the Christian religion represents that she is dependent on a determined contentual communication that must be appropriated in an understanding manner (*verstehend*) and articulated symbolically as religion.

God thus comes in the Christian religion from God through God as God.[16] God becomes real in the Christian religion as the Christian religion itself becomes real. Only in this way is the word "God", which is bound to the memory of Jesus Christ and passed on in culture, appropriated in an understanding manner (*verstehend angeeignet*) as a religion by human beings and used to articulate their religion. In the Christian religion, God represents the fact that the Christian religion arises underivably from the communicated content and has its foundation, validity, and truth in itself. God is an image of the Christian religion as religion. He is not simply an object like other objects, but such an object by means of which the function of the objects of the Christian religious communication becomes illustrative for them to be intentionally used in a religious and not in a cultural manner. Thus, it is clear that statements about God cannot be factual statements about an object. Rather, all religious statements have a reflexive function. They describe the reflexive structure of the Christian religious communication.

With the derivation of the Christian religious idea of God and its function for the Christian religion, the systematic foundations have been outlined to such an extent that God's relation to evil can now be discussed.

## 4. God and Evil in a Theology of Christian Religious Communication

With its idea of God, the Christian religion presents itself as a transparent, self-referential, and structured communication event. In this sense, God is not a concept that refers to a given object about which statements must be produced, but an index for the Christian religion itself (cf. Dalferth 1992; Wittekind 2018, p. 89). From the religious idea of God outlined so far, a thematization of evils emerges that opens a new perspective compared to the theodicy debates presented in the second section. For if, as explained, religious statements about God cannot be understood as factual statements about an object, then the question of theodicy, that is, whether, and, if so, how, the three statements—(a)

God is good, (b) God is omnipotent, and (c) there is evil in the world—can be true together, is misguided from the outset. Statements about God do not have a representational function but a reflexive one. They describe the reflexive structure of the Christian religious communication.[17] In the considerations that follow, the attributes of God must be briefly discussed on the basis of the religious idea of God elaborated in the third section, so that the question of how evil occurs in the Christian religion can then be investigated.

The classical form of the doctrine of the attributes of God as well as the substance-metaphysical version of the idea of God on which it is based can no longer be continued under the critical epistemological conditions of modernity. With this, the distinction of essence and attributes of God, which is constitutive for the classical doctrine of God, is also dropped. This distinction is comprehensible only under the assumption of the Aristotelian metaphysics of substance. Consequently, the attributes of God are not something that is added to a given essence, so that the question arises how these attributes can consistently coexist in the essence of God. Rather, the essence and attributes of God have a function for the reflexive description of the Christian religion.[18] Only in this way does the religious function of the idea of God become clear, which distinguishes it from a philosophical concept of God or a principle of world explanation. The Trinitarian God is a reflexive descriptive element in the Christian religion, with which the Christian religion represents its own functioning as religion in the use of contents in communication.

How must the essence and attributes of God be understood in a theology of the Christian religious communication? The starting point is the classical dogmatic determination of God as *essentia spiritualis infinitia*. This determination, however, does not establish a metaphysical object to which it refers, but has a reflexive function. It describes the Christian religion as an autonomous form in culture that arises in the religious use of content in communication and knows about the religious intention of this content. The reflexive self-transparency and self-referentiality, in which the Christian religion exists in the use of contents, is represented in its idea of God. God's absoluteness and transcendence are descriptive elements with which the Christian religious communication depicts both its origin, which cannot be derived from the communicated content, and its existence in the religious use of this content. Consequently, absoluteness is not a feature of content, but an expression of the self-relationship of the Christian religion.

Similar to the Trinitarian God, his attributes must not be understood in a representational sense. They explicate the reflexive structure of the Christian religious communication in the use of contents. This is the parallel between the doctrine of the attributes and the doctrine of the Trinity (cf. Barth 1948, p. 367). However, unlike the latter, the doctrine of attributes does not explicate the structural elements of content, appropriation, and articulation, from whose interrelation the Christian religion emerges, but rather their reflexivity in the use of content in communication. God is not simply a representational content in the Christian religion, but a content that gives expression to reflexivity in the use of content in religious communication (cf. Wittekind 2018, p. 92). The dogmatic doctrinal tradition distinguished two sets of attributes of God: attributes that belong to God absolutely (*attributa absoluta*) and attributes that belong to him in his relation to the world (*attributa relativa*). This distinction is taken up here in such a way that the absolute attributes of God are related to the doctrine of God in the narrower sense and the relative ones to God's relationship to the world in the horizon of the doctrine of creation and providence.

If the attributes of God represent forms of description of the successful reflexive use of contents in the Christian religious communication, then, on the level of the doctrine of God in the narrower sense, they explicate the independence, non-justifiability, and inner functioning of the Christian religion. The unity, immutability, and infinity of God describe the transparency and self-referentiality of the Christian religion that establishes itself in communication, which is not derivable from the world, that is, from the contents of communication, and functions transparently as an autonomous form of communication in culture. While the absolute attributes of God function as descriptive elements of the transparent use of contents in the Christian religious communication, the world-related attributes of

God are concerned with the functioning of the Christian religious communication on the concrete contents, i.e., with the inclusion of the world in the Christian religion. Both forms of the attributes cannot be separated, since in each case it is God himself who comes up in the absolute and the relative attributes. Their difference lies solely in the fact that in the doctrine of creation and providence the transparent and self-referential functioning of the Christian religion represented by the idea of God is transferred to the world in the religious use of contents. God comes into the world only in the Christian religious communication, in that one's own life in the world is included in the Christian religion. Additionally, it is only here, in the inclusion of the world in the Christian religion, the problem of evil becomes virulent. It presupposes the creation of the world and is treated, in the structure of theological dogmatics, in the doctrine of providence, which forms a part of the doctrine of creation.

The creation statements of the Christian religion are also understood within the framework of a theology of Christian religious communication not in representational terms but as reflexive forms of description of the Christian religious communication (cf. Danz 2021a, pp. 1–7; Wittekind 2018, pp. 115–32). The doctrine of creation is concerned with the fact that everything in the world can become an object of religious communication and, in this way, be included in the Christian religion.[19] Faith in creation, therefore, does not thematize the world as such or provide an explanation of its origin. It describes the world as it appears in the Christian religion. However, the inclusion of the world in the Christian religious communication does not depend on characteristics or particularity of the world that qualify it for this. Everything in the world can be included in the Christian religion and adopted as its object. In contrast to the doctrine of creation, the doctrine of providence relates God to the life of the individual in the world. Therefore, the doctrine of providence is no longer concerned with the fact that everything in the world can become the object of the Christian religion, but rather with the application of the Christian religion to the concrete events in life. Now, what does this mean for the relationship between God and evil?

God comes to reality in the Christian religious communication. This must be constantly re-established by including the concrete events that happen to the life of a person in the Christian religion. The Christian religion depends on people's religious use of the contents in communication. If the Christian religious communication succeeds in the concrete events of life, then the Christian religion arises, which is represented in the idea of God. God is then transferred to the world and the concrete events in it. By succeeding at the concrete events of life, the Christian religious communication cannot be questioned by them. Since the Christian religion cannot be derived from contents, the nature or quality of these contents are irrelevant. By incorporating concrete events from the world into the Christian religion, they no longer have cultural or ethical significance, but become an expression of the Christian religion. This also applies to the evils that befall a person in their life. If they are included in the Christian religion, they become subject to God's power and become the object of praise and lamentation to God. God's omnipotence, similar to God's goodness, is not a representational attribute that belongs to an object. It describes the transparent functioning of the Christian religious communication based on concrete contents of communication. This has its justification, truth, and validity in itself, not in determined experiences. Thus, neither the omnipotence of God nor his goodness can be refuted by events in the world, be they positive or negative.

However, since God only comes into the world if the Christian religious communication is successful, and this communication must be constantly re-established based on concrete events in the world, there is always also the possibility that the communication does not succeed. Then, concrete experiences of evils are not related to God, because the Christian religious communication fails at them. This does not falsify God either, since there is always the possibility of interpreting experiences of evil and good in a non-religious way. The Christian religion is, as explained, not an explanation of the world, but its own form of communication besides other cultural modes of communication. Its objects come to

existence only in the Christian religion and are not given outside of it. Since religion is not an anthropological necessity, not all events in the world have to be interpreted religiously.

God's world-related attributes such as omnipotence and goodness describe, as has been shown, the transparent functioning of the Christian religion in the concrete contents of life. Omnipotence and goodness have a reflexive not a representational function. Since the reality of God in the Christian religion depends on the success of the Christian religious communication in the concrete events of life, and this success can neither be derived nor justified, the possibility of failure of this communication always remains. What does this mean for the problem of theodicy? In the first place, it is not a theoretical-logical problem that can be solved intellectually. In the second place, against the background of the outlined considerations on the function of the idea of God in the Christian religion, the theodicy problem and the different answers given to it can be understood as an abstract echo of the success or failure of the inclusion of concrete experiences in the Christian religion. However, the theodicy debate raises the success or failure of the Christian religious communication to a general logical level by abstracting it from the self-view of the Christian religion and reformulating it as a question about the possible truth of the three propositions: (a) God is good, (b) God is omnipotent, and (c) there is evil in the world. On this level, however, the relationship between God and evil cannot be resolved.

**Funding:** This research received no external funding.

**Data Availability Statement:** Not applicable.

**Acknowledgments:** I thank Fábio Henricque Abreu (Rio de Janeiro) for help with the English translation of the article.

**Conflicts of Interest:** The author declares no conflict of interest.

## Notes

[1] Cf. Kant (1983, p. 105): "By a theodicy is meant the defense of the supreme wisdom of the world's author against the charge that reason brings against it from what is contrary to purpose in the world."

[2] Thus Hans Jonas suggested to renounce the predicate of God's omnipotence and to hold on to that of goodness. Only in this way, against the background of the Shoah, the idea of God could be held on to. Cf. Jonas (1987).

[3] Provisions that, following Augustine, understand evil as *privatio boni*, amount to an abolition of evil.

[4] Cf. Mackie (1990, p. 26): "However, the contradiction does not arise immediately; to show it we need some additional premises, or perhaps some quasi-logical rules concerning the terms 'good', 'evil', and 'omnipotent'. These additional principles are that good is opposed to evil, in such a way that a good thing always eliminates evil as far as it can, and that there are no limits to what an omnipotent thing can do."

[5] Cf. Mackie (1990, p. 26): "From these it follows that a good omnipotent thing eliminates evil completely, and then the propositions that good omnipotent things exist, and that evil exist, are incompatible."

[6] Cf. Swinburne (1987, p. 290): "If God wants to give man the opportunity both to acquire knowledge and to determine his own destiny, he can only do so by giving him the opportunity to acquire knowledge in the normal inductive way." On Swinburne's understanding of induction, cf. ibid., pp. 277–88.

[7] Cf. Swinburne (1987, p. 294): "Assuming, then, that the world owes itself to no morally reprehensible act of creation, there must be evils of various kinds in it if such behaviors as courage, compassion, etc. are to be possible. Such evils give human being the chance to realize the highest virtues." For Swinburne, Hiroshima and Bergen-Belsen (cf. ibid., p. 301) are also evils that promote a higher good. For a critique of such functionalizations of evil, cf. also Phillips (2005, pp. 49–94).

[8] Swinburne's solution to the theodicy problem thus does not go beyond what Mackie calls fallacious solutions. Cf. Mackie (1990, pp. 27–32).

[9] In this connection, cf. the proposal advanced by Friedrich Hermanni, who based on Leibniz, understands the evils as *logically* necessary components of the world created by God. Unlike Swinburne, Hermanni includes Mackie's two additional assumptions in his argumentation and distinguishes between a logical and an empirical theodicy problem. For him, the logical theodicy problem can be resolved solely by assuming "that the evils are not prevented by an omnipotent and omnibenevolent God because they are logically necessary elements of the unsurpassable good world he created" Hermanni (2009, pp. 16–21). Cf. also Hermanni (2002).

[10] Laura Garcia's proposal does not get beyond this dilemma either cf. Garcia (2017, pp. 57–89). God, she argues, does not cooperate in evil actions because, due to his perfect goodness, he does not share the evil intention of the action. God, since he has an effect

on everything, including evil actions, only creates their conditions and allows them to happen. This model works only if one accepts the Thomistic doctrine of the two causes. However, apart from the fact that the distinction between a first and a second cause in actions cannot be maintained, it is impossible to see how a finite causality of action can exist alongside an infinite one.

[11] In the more recent debate, therefore, the proposal has been made to combine the logical theodicy problem with an empirical one, i.e., to include the self-view of the sufferers in the debate. Cf. Hermanni (2009, pp. 16–21), Klinge (2019, pp. 165–83). However, since at the same time a metaphysical-theistic concept of God is held on to, even these extensions do not arrive at an appropriate way of dealing with the theodicy problem.

[12] Friedrich Schleiermacher's dogmatics, *Der christliche* Glaube (1821/22; 2nd ed., 1830/31), is fundamental for this religious-theoretical reshaping of scientific theology.

[13] Thus, a general concept of religion is dispensed with. By limiting the theological concept of religion to Christianity, the possibility is opened to recognize in theology that other religions already understand what religion is differently than Christianity. It is thus a matter of a pluralization of the understandings of religion. Cf. Danz (2020, pp. 101–13).

[14] The Bible represents, in the Christian religion, the dependence on a determined contentual communication as memory of Jesus Christ.

[15] This circle is explicated by the theological concept of revelation. Cf. Wittekind (2018, pp. 89–90), Danz (2022, pp. 601–26).

[16] On this formula, cf. Jüngel (1992, pp. 521–34). In contrast to Jüngel, who constructs the Trinitarian God as the presupposition and foundation of the Christian religion of faith, here the doctrine of the Trinity is used as an explication of the self-referential structure of the Christian religion.

[17] This was already the proposal of Friedrich Schleiermacher and his reformulation of the classical doctrine of properties against the background of modern epistemological criticism. Cf. Schleiermacher (1999, p. 254), § 50 leading sentence: "All the properties which we attribute to God should not designate anything special in God, but only something special in the way of relating the feeling of absolute dependence [*schlechthinniges Abhängigkeitsgefühl*] to him." Schleiermacher's redetermination of the attributes of God as structural descriptions of the religious act has been followed by the further development of the doctrine of God in the Protestant dogmatics. In contrast to Schleiermacher and 19th century theology, however, 20th-century Protestant theology no longer based the idea of God on a general concept of religion already anchored in the structure of consciousness, but elaborated the idea of God as a theological description of the religious act that, without anthropological presuppositions, originates underivably in human beings. Thus, the doctrine of the attributes of God unfolds the reflexive structure of the self-referential revelation of God in the act of faith. Cf. Barth (1948, pp. 362–764).

[18] In this sense, the doctrine of properties is consistently constructed in the doctrine of God in recent Protestant dogmatics. Cf. Barth (1948, p. 383), Weber (1964, pp. 463–64), Härle (2000, pp. 255–56).

[19] Thus, the soteriological interpretation of the faith in creation as an extension of the faith in salvation is taken up and continued.

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
