# Peer review of "God and Evil—Systematic-Theological Reflections on the Doctrine of God"

_religions, doi:10.3390/rel13111075_

Round 1
Reviewer 1 Report
This article is significant in terms of positing Christian theology as religious communication. The article succeeds then to argue that the question of evil is not only specific to Christian theology/Christianity as a religion. It correctly argues that evil is a reality in the world. It is correct then as the article argues to address evil but with the view that failure to address evil does not negate God, His goodness and omnipotence. I believe to boost the author's claim that God becomes a reality through Christian religious communication, I believe narrative theology would be a good partner to substantiate this claim as the narrative theologians argue that the message and meaning of the Bible or its theology is within the Bible itself. However, what is troubling about this article is that there is not an adequate account for the identity and nature of God, which we know through God's revelation that is not reliant on us. God simply seems to be fluid as opposed to a true reality of the transcendent. Also, it would be important for the authors to accept the fact that Christianity theology is in fact trapped in western dualisms of having to choose either the extreme of either or, which might not be true for the world of the Bible. At the same time these dualism have their own approach to truth through analytic philosophy or syllogisms. I would argue that grey areas even though they may not fit our conception of truth, logic and facts affirm the mystery in the world and the mystery of God. Furthermore, the scholars must be clear and concrete about the evil they are referring to because evil must not be treated as an abstraction (there are agencies of either good or evil). Certainly, natural events should not be linked with human morality because evolution and creation is complex.
Author Response
Thank you for the comments on the article, the point of which is indeed to outline a new approach in the doctrine of God, with which the alternative reality assertion of God or suspicion of fiction is to be overcome. This happens explicitly not in a narrative theology, but in a theology of Christian-religious communication. In the becoming-real of religious communication God becomes real, namely as an event of faith. From here, as explained, a new view of the problem of theodicy arises.
Reviewer 2 Report
The paper is interesting, though not very original. The idea that statements about God are neither factual nor representational is a commonplace, especially in the context of German theology, which the author heavily relies on. The application of this idea to the problem of evil, however, is intriguing. Nevertheless, simply relying on the self-view of the religious subject seems a little too easy. In the second part of his/her paper, the author refers to the "recent theodicy discourse" by discussing Mackie and Swinburne. I would recommend including some more recent authors (especially from the analytical tradition of the philosophy of religion), since the almost classical arguments of Mackie and Swinburne do not represent the current state of debate.
Author Response
Thank you for the comments on the article, which is indeed in the context of the German-language theological debate. But this is carried on at central points by the idea of a theology of Christian-religious communication. It is not simply a matter of adopting the self-view of the Christian religion, but of constructing its transparent functioning as a religion. This construction is comprehensible in general and without special presuppositions.
The positions of Mackie and Swinburne serve only as a starting point to structure the more recent debate, which is included by means of the positions of Sterba, Jordan, Garcia, and others. The function of the section on recent theodicy debates is to show their aporia: of the three statements, one is dropped. Even if more recent literature were included, the result would be the same. For this reason, I have referred to a few exemplary contemporary positions.